# The Role of p53 Mutations in Early and Late Response to Mitotic Aberrations

**DOI:** 10.3390/biom15020244

**Published:** 2025-02-08

**Authors:** Anna Hertel, Zuzana Storchová

**Affiliations:** Group Molecular Genetics, Faculty of Biology, RPTU Kaiserslautern-Landau, Paul Ehrlich Str. 24, 67663 Kaiserslautern, Germany

**Keywords:** *TP53*, chromosome missegregation, whole-genome doubling, aneuploidy, chromosomal instability

## Abstract

Mutations in the *TP53* gene and chromosomal instability (CIN) are two of the most common alterations in cancer. CIN, marked by changes in chromosome numbers and structure, drives tumor development, but is poorly tolerated in healthy cells, where developmental and tissue homeostasis mechanisms typically eliminate cells with chromosomal abnormalities. Mechanisms that allow cancer cells to acquire and adapt to CIN remain largely unknown. Tumor suppressor protein p53, often referred to as the “guardian of the genome”, plays a critical role in maintaining genomic stability. In cancer, CIN strongly correlates with *TP53* mutations, and recent studies suggest that p53 prevents the propagation of cells with abnormal karyotypes arising from mitotic errors. Furthermore, p53 dysfunction is frequent in cells that underwent whole-genome doubling (WGD), a process that facilitates CIN onset, promotes aneuploidy tolerance, and is associated with poor patient prognosis across multiple cancer types. This review summarizes current insights into p53’s role in protecting cells from chromosome copy number alterations and discusses the implications of its dysfunction for the adaption and propagation of cancer cells.

## 1. Introduction

The *TP53* gene was discovered by several research teams in 1979 and was initially thought to function as an oncogene because of its frequent expression in cancer cells. However, in the late 1980s, its true role as a crucial tumor suppressor was identified [1]. The p53 transcription factor encoded by *TP53* is essential for cancer prevention, as it regulates cell cycle checkpoints, promotes DNA repair, maintains genomic stability, and triggers apoptosis in response to cellular stress [2]. Mutations in *TP53* that disrupt those protective pathways are among the most common genetic alterations in cancer [3].

Years of research established p53 as the “guardian of the genome”, highlighting its critical role in maintaining genome integrity and preventing genomic instability, which is characterized by increased alterations in DNA sequence, structure, or number. The changes in chromosome copy number, either aneuploidy, which refers to an unbalanced number of chromosomes, or whole-genome duplications (WGD), are usually associated with chromosomal instability (CIN). CIN describes the ongoing process of chromosome missegregation that leads to variability in chromosome numbers over time. These processes play a crucial role in the initiation and progression of cancer, and p53 mutations facilitate this process. This link is well documented in both human cancer samples and mouse models; the inactivation of p53 predisposes cells to rapid tumor development, underscoring its fundamental role as a tumor suppressor [4,5]. Interestingly, the loss of functional p53 triggers a predictable pattern of cancer genome evolution, which includes deletions, genome doubling, and amplifications [6]. This suggests that there is an active interplay between p53 functionality and the maintenance of genomic stability, and that mutations in p53 profoundly influence the evolution of the cancer genome, shaping both the degree and type of aneuploidy.

Multiple signaling pathways are activated by p53 in response to a variety of cellular stresses, such as DNA damage, oncogene activation, hypoxia, and cellular energy metabolism. The loss of p53 impairs not only genomic stability but also disrupts mitochondrial function and alters metabolism, resulting in decreased oxygen consumption, impaired mitochondrial respiration, and increased glucose utilization [7,8]. These factors interfere with the cellular homeostatic mechanisms that govern DNA replication, chromosome segregation, and the accuracy of cell division. To exert its varied effects, p53 binds to DNA in a sequence-specific manner, regulating the expression of target genes [7,9]. According to the cell type and the level of stress the cell is exposed to, p53 may induce cell cycle arrest, apoptosis, or senescence. In this manner, damaged cells are prevented from propagating and potentially causing cancer [7].

This review explores the connection between p53 mutations and chromosomal instability, focusing on two key aspects: the early cellular response to chromosome missegregation and WGD, and the later adaptive response to CIN and aneuploidy. It highlights how these processes contribute to tumor heterogeneity and promote cancer progression. By examining the dynamic interplay, the review aims to identify potential therapeutic targets for mitigating the effects of CIN in p53-mutant cancers.

## 2. The Function of p53

As a transcription factor, p53 primarily exerts its effects by regulating the expression of target genes [10]. p53 targets drive protein synthesis for DNA repair, reducing DNA damage [8,11], as well as enhancing endosomal activity, which helps to remove growth receptors from the cell surface and release exosomes to alert the immune system to cellular stress [12,13]. Activated p53 inhibits cell growth by targeting IGF-1/AKT and mTOR pathways, indirectly suppressing mTORC1 via AMPK activation and upregulating *REDD1*, *PTEN*, *TSC2*, and Sestrins, thus maintaining proliferation control and homeostasis [14,15,16]. Furthermore, p53 regulates mitochondrial respiration by controlling genes such as *GLS2*, which is involved in antioxidant defense and energy metabolism, and *TIGAR*, a glycolysis regulator [7,17,18,19]. Additionally, p53 plays a central role in apoptosis by transcriptionally activating pro-apoptotic genes such as *BAX*, *PUMA*, and *NOXA*, while suppressing anti-apoptotic genes including *BCL-2*, thereby orchestrating programmed cell death [20]. Thus, p53 serves as a multifaceted regulator of cellular processes, orchestrating DNA repair, cell cycle arrest, apoptosis, and metabolism through its interactions with diverse pathways and target genes, thereby maintaining cellular homeostasis and preventing tumorigenesis.

### The Canonical Model of p53 Activation

The abundance and stability of p53 are critical for its activity. The regulation of cellular p53 levels mainly occurs through ubiquitin-mediated proteasomal degradation. In an unstressed cell, ubiquitination by the p53-specific E3 ligase, MDM2, targets p53 for proteasomal degradation. Upon stress, the phosphorylation of p53 disrupts the interaction with Mdm2, which leads to the stabilization of p53 (Figure 1) [21,22,23]. Various kinases, including the DNA damage response kinases ATM/ATR/DNA-PK and CHK1/CHK2, mediate this modification following DNA damage, particularly at serine residues Ser15 and Ser20 [24,25]. Rapid phosphorylation of these N-terminal regulatory sites activates p53 in response to stress signals. It should be noted, however, that p53 can be activated without phosphorylation [26].

The second step in the canonical model involves the binding of p53 to specific response elements in target gene promoters (Figure 1). This binding facilitates chromatin remodeling through the direct interaction of p53 with specific histone modifiers or their associated proteins, helping to recruit histone transacetylases and methyltransferases. Beyond merely recruiting these modifiers, p53 actively participates in reorganizing chromatin architecture in response to stress. During p53 activation, p53-bound enhancers are activated, marked by increased H3K27ac and elevated levels of nascent RNA transcription, amplifying the transcriptional response (Figure 1). Cohesin-mediated DNA looping facilitates interactions between p53-bound enhancers and gene promoters, enabling the propagation of p53’s regulatory effects and driving a coordinated transcriptional response [27,28,29,30,31]. While primarily a transcriptional activator, p53 also suppresses certain genes through sequence-specific DNA binding [32]. Given its critical role in regulating both activation and suppression of target genes, it is not surprising that most p53 mutations in cancers impair or completely abolish its DNA-binding ability [33].

## 3. *TP53* Mutations

Activated p53 forms a homotetramer. Each monomer contains several domains, including an N-terminal transactivation domain, a proline-rich region, a conserved DNA-binding core domain, a C-terminal tetramerization domain, and an unstructured basic domain (Figure 2A) [34,35]. p53 mutations in cancers are primarily somatic and can be caused by environmental factors or random replication errors [36]. However, heterozygous germline mutations in the *TP53* gene can lead to a hereditary condition known as Li-Fraumeni syndrome (LFS). LFS is marked by an early cancer onset, including sarcomas, breast cancer, and brain tumors, due to an inherited *TP53* mutation that leaves only one functional allele, which is often lost through a second somatic mutation, resulting in complete p53 inactivation in tumor cells [37,38].

The functional consequences of p53 mutations, whether somatic or inherited, are diverse and critical to cancer progression. Some p53 mutations result in a dominant-negative effect (DNE), where the mutant p53 protein encoded by the mutated allele interferes with the wild-type p53, impairing its tumor-suppressing function. This is typical in the DNA-binding domain, where the majority of *TP53* mutations occur. As mutations accumulate with age, loss of heterozygosity (LOH) often follows initial heterozygous mutations, resulting in the complete loss of functional p53. LOH can be a bi-directional process, but usually the wild-type allele is lost [39]. Biallelic *TP53* mutations, including mutations either in both alleles or a deletion in one and a mutation in the other, are frequent in most aggressive cancers. These mutations lead to complete *TP53* inactivation and correlate with poor prognosis [38,40,41]. The specific type and context of p53 mutations or disruptions can exert distinct functional effects, leading to diverse consequences on cellular processes and influencing tumor behavior.

### 3.1. Missense Mutations

The most common type of *TP53* mutations are missense mutations, occurring in approximately 87% of *TP53*-mutated cancers [42,43]. These mutations typically result in a single amino acid substitution within six “hotspot” codons—175, 245, 248, 249, 273, and 282—of the DNA-binding region (amino acids 102–292), impairing its ability to regulate the target genes (Figure 2). These “hotspot” mutations account for nearly 30% of all *TP53* mutations and may disrupt the overall tertiary structure of p53 (e.g., R175, G245, R249, and R282), or impair p53’s ability to bind DNA (e.g., R248 and R273) (Table 1) [42,44]. By disrupting p53’s binding to DNA and the activation of genes such as *CDKN1A*, *BAX*, and *GADD45*, these mutations contribute to uncontrolled cell proliferation, the evasion of apoptosis, genomic instability, and, in some cases (e.g., R175H and R248Q), more aggressive tumor phenotypes, increased metastatic potential, and chemotherapy resistance [9,45,46].

### 3.2. Gain of Function Mutation

Mutated p53 may also exert a pro-oncogenic function through gain-of-function (GOF) alterations. The GOF p53 mutant proteins generally do not bind DNA directly, but can bind indirectly through interactions with other transcription factors and protein effectors. The common hotspot sites R175 and R273, frequently found in cancers, have been identified to possess GOF properties, but not all missense mutations exhibit GOF [61,62]. GOF mutations also enhance oncogenic functions via a cytosolic DNA response induced by CIN, which activates the cGAS-STING pathway and promotes chronic inflammation, contributing to tumor progression and metastasis [61]. Additionally, GOF p53 mutants disrupt normal cell cycle regulation [17]. Genome-wide analyses have shown that GOF p53 can associate via other proteins with the promoters of genes encoding cyclin A and CHK1, activating their transcription to support DNA replication. Furthermore, they may inhibit the starvation response regulator AMP kinase (AMPK), disrupting its role in mitigating oxidative stress, which contributes to elevated ROS levels in tumor cells [63,64]. Importantly, GOF p53 allows the accumulation of cells with aneuploid karyotypes [63].

### 3.3. TP53 Mutation Dominant-Negative Effect

Some mutations result in a dominant-negative effect (DNE), which occurs when mutant p53 and wild-type p53 are expressed in a cell. Unlike most tumor suppressors, where the loss of both alleles is required, a single p53 DN mutation can promote cancer progression. The DNE depends on physical interactions between wild-type and mutant p53 through their intact tetramerization domains [65,66]. A co-expressed mutated form of p53 may be incorporated into the tetramer and potentially impair its tumor suppressor activity by reducing its affinity for DNA. Studies in transgenic mice and patients with LFS show that dominant-negative p53 mutations are associated with higher tumor incidence and earlier onset, highlighting their aggressive role in oncogenesis [67].

The key difference between p53 GOF and DNE mutations lies in how the pro-oncogenic phenotype is observed: DNE occurs when a mutant p53 allele, in a heterozygous state (p53Mut/WT), exhibits a pro-oncogenic phenotype that is not present in heterozygous p53+/− cells. In contrast, GOF is identified when the pro-oncogenic phenotype is observed in p53Mut/− cells, but absent in p53-null (p53−/−) counterparts. However, distinguishing between the two can be complex. GOF can still manifest even in the presence of wild-type p53, meaning that simply demonstrating DNE does not rule out the possibility of GOF. Additionally, in some tumors, the wild-type p53 allele may be lost through LOH, which complicates the analysis [68,69]. To fully exclude or validate GOF, one must compare the phenotypes of hemizygous mutant p53 (p53Mut/−) and p53 knockout, as well as tumors with and without LOH, ensuring a clear distinction between the two effects.

### 3.4. The Complexity of Gain-of-Function p53 Mutations

The idea that mutant p53 exhibits GOF properties was proposed nearly three decades ago. Indeed, certain *TP53* mutants, such as R175H and R273H, drive the formation of tumors with more aggressive and invasive profiles compared to tumors lacking p53 entirely, providing strong evidence for a GOF effect [70]. Additional studies have demonstrated that GOF mutations enhance metastasis, stemness, and the epithelial-to-mesenchymal transition. These behaviors are also associated with aneuploidy and CIN, and arise due to altered gene expression patterns caused by imbalanced chromosome numbers (Figure 3) and disrupted regulatory networks, following WGD, which is frequently observed in cancers (Figure 2B) [71,72].

Further support for the GOF hypothesis comes from clinical observations that patients with germline *TP53* missense mutations often develop cancer earlier than those with p53 loss [73]. Similarly, mice with certain p53 mutations develop more aggressive pleomorphic rhabdomyosarcoma tumors than their p53-null or wild-type counterparts [74]. The GOF effects are likely tissue-specific; for example, when similar p53 mutations were introduced into lung tissue, no noticeable gain-of-function activity was detected compared to p53 loss [75].

Thus far, there is no unified model to explain the diverse GOF effects seen across different mutations. The variable behavior of individual mutations is challenging to address, especially since studies often only focus on a limited number of mutations. For instance, while R172H and R270H (analogous to human R175H and R273H) demonstrate GOF activity in mouse models [70], the G245S and R249S mutants do not exhibit GOF, while R246S acts in a dominant-negative way, promoting cell survival after radiation [76,77]. R248Q knock-in mouse models show earlier tumor onset and shorter lifespan compared to p53-null mice, an effect not observed with R248W, despite both being structural mutations [78]. Notably, LFS patients with R248Q p53 develop cancer earlier than those with G245S p53 or p53-null mutations, highlighting the unique role of R248Q (Table 1) [76].

Remarkably, a recent study indicated that such effects may be more accurately attributed to aneuploidy arising in the mutant cells rather than the mutations themselves. By demonstrating that GOF phenotypes only manifest in aneuploid cells, the study suggests that previous interpretations attributing oncogenic properties solely to mutant p53 may need reevaluation and reinforces the idea that chromosomal alterations play a crucial role in determining cancer phenotypes [79]. These findings highlight the need for further research to clarify the roles of GOF p53 mutations and their link to chromosomal alterations in understanding how mutant p53 contributes to cancer progression and therapy resistance.

### 3.5. Isoforms, Frameshift, or Splice Mutations of TP53

*TP53* diversity is further expanded by the existence of 9 known p53 mRNAs in human cells, which encode 12 different p53 protein isoforms, each expressed in a tissue-dependent manner [80]. Numerous *TP53* isoforms can be produced using alternative splicing or a different start codon (Figure 2), suggesting that some mutations may only target some isoforms, leaving the remaining isoforms intact [80,81]. Abnormal expression of these p53 isoforms has been observed in breast, colon, ovarian, and lung cancers, highlighting their potential role in cancer progression [80,81,82,83,84]. Indeed, functional studies of the heterozygous *TP53*β-stop-lost variant affecting the p53β isoform reveal that this variant extends the p53β isoform, enhancing its interaction with the canonical p53 and potentially disrupting normal p53 signaling pathways [85]. These findings emphasize the need to consider alternative splicing and isoform expression when assessing the role of p53 in cancer. Thus, cancer development can be linked to silent *TP53* mutations or mutations in non-coding regions, such as introns or splicing sites, probably because they cause unbalanced expression of p53 isoforms despite expressing the wild-type p53 protein [83,86].

**Figure 2 biomolecules-15-00244-f002:**
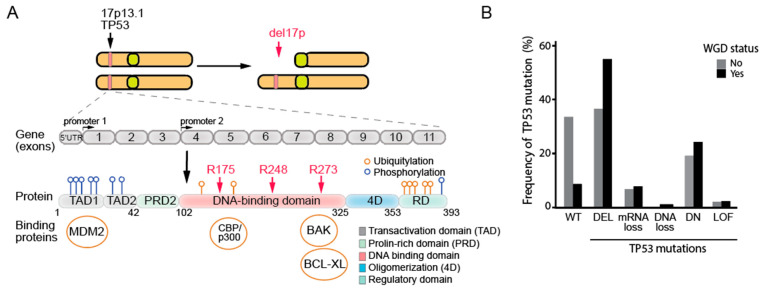
Structure and mutations of *TP53*. (**A**) Human *TP53* located on chromosome 17 codes for a protein composed of five main domains: the amino-terminal transcription activation domain (TAD), the DNA binding domain (DBD), the proline-rich domain (PRD), the regulatory domain (RD), and the oligomerization domain (4D). A common cause of *TP53* loss is the del(17p). Mutations in *TP53*, which include missense, truncating, in-frame, and splice mutations, arise from various mechanisms and contribute to tumor development. The p53 core domain, crucial for DNA binding, contains common mutation hotspots at residues R175, R248, and R273. Many post-transcriptional modifications and interaction pathways further regulate the p53 function and its interaction with binding proteins. The depiction is based on data from Chen et al. (2022) [87] and Buter and Amelio [88]. (**B**) Frequency of *TP53* mutation types by WGD status. The frequency of *TP53* mutations was calculated using the available data from the DepMap Portal (Release DepMap 24Q4 Public) [89]. The data sets were filtered to include only No WGD and 1 × WGD cases, while excluding 2 × WGD and NA (not available) values. The tumor suppressor status of *TP53* across the cancer cell lines was extracted from Sonkin et al. (2013) [90] and mapped to the DepMap data. Notably, in 5–8% of cancers, WGD occurs independently of p53. The p53 pathway may be disrupted in these cells by other defects, despite p53 itself remaining intact [91]. (WT = wild-type, DEL = deletion of one allele, DN = dominant negative, LOF = loss of function).

**Figure 3 biomolecules-15-00244-f003:**
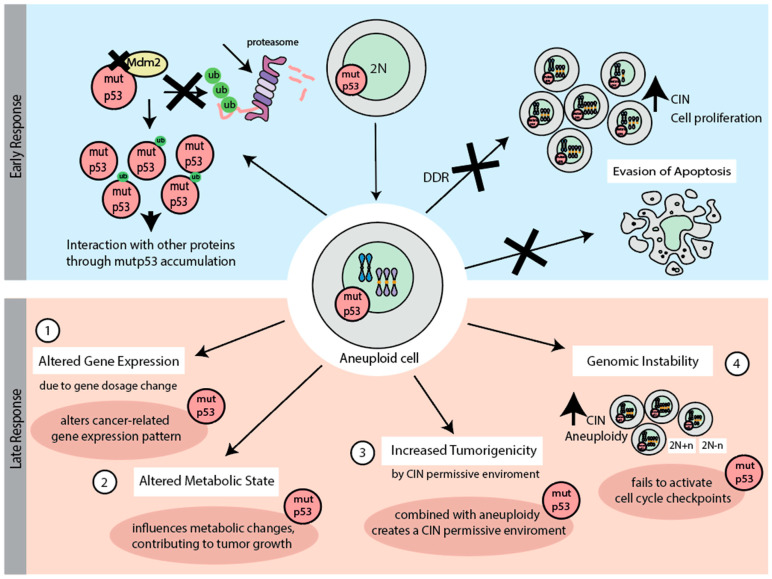
The role of mutant p53 (mutp53) in cellular responses to chromosome missegregation, divided into early and late phases. In the early response, mutp53 accumulation results from impaired proteasomal degradation, allowing interactions with other proteins. This leads to tolerance of chromosome missegregation by suppressing the DNA damage response (DDR) and apoptosis, enabling continued cell proliferation. In the late response, (1) mutp53 alters transcriptional programs and gene expression, though not directly linked to gain-of-function (GOF) effects, and (2) supports metabolic changes driven by gene dosage imbalances, facilitating aneuploid cell survival and tumor growth. (3) Mutp53 in combination with aneuploidy creates a permissive environment for cancer development. (4) Additionally, mutp53 contributes to genomic instability by failing to activate cell cycle checkpoints, promoting continued chromosomal instability (CIN) and tumor progression.

### 3.6. The Deletion of the 17p Arm and Its Relevance to the Loss of p53 in Human Cells

The deletion of the 17p arm (del17p), where the *TP53* gene is located, is a large genetic alteration frequently observed in various human cancers, leading to the loss of not only p53 but also the reduced expression of hundreds of other genes, with the possible adverse risk of haploinsufficiency [92,93]. 17p deletion is driven by CIN and often leads to the loss of functional p53, as it is typically combined with a mutation in one p53 allele [94,95].

Interestingly, a recent study highlighted the prevalence of chromosome 17 in micronuclei (MN), particularly in triple-negative breast cancer cells, suggesting that CIN disproportionately impacts this chromosome, possibly due to its gene-dense structure, including tumor suppressors such as *TP53* and *BRCA1*. The study further proposed that the selective sequestration of chromosome 17 into MN may represent a mechanism for tumor cells to bypass cell cycle checkpoints or discard tumor suppressor genes such as *TP53*, which could exacerbate genomic instability and tumor progression [96]. Clinically, the prognostic significance of deletion 17p, which involves the *TP53* gene, has been extensively studied and has been well documented in the context of several malignancies, including acute myeloid leukemia (AML) and non-small cell lung cancer (NSCLC). In these cancers, the presence of 17p deletions is associated with poor clinical outcomes, increased tumor aggressiveness, resistance to conventional therapies, and a higher likelihood of relapse [97,98,99]. Beyond AML and NSCLC, the prognostic and predictive value of *TP53* deletions or mutations has been increasingly recognized in chronic lymphocytic leukemia (CLL). *TP53* abnormalities in CLL are linked to poor survival and therapeutic resistance, underscoring the importance of testing for *TP53* deletions and mutations prior to initiating treatment, as this can guide clinical decision-making [100,101,102].

While *TP53* loss plays a decisive role in tumor initiation and progression, it remains uncertain whether other genes within the deleted region contribute to tumorigenesis beyond the loss of *TP53* alone. A recent study highlighted that the loss of EIF5A and ALOX15B, located in the syntenic region of mouse chromosome 11B3 (analogous to human 17p13.1), cooperates with *Trp53* deletion to drive more aggressive disease in lymphoma and leukemia, suggesting that additional genes in the 17p region may have critical roles in cancer progression [103].

## 4. *TP53* Mutations Are Strongly Linked to Aneuploidy and Whole-Genome Doubling in Cancer Genomes

The interplay between p53 alterations and CIN has been extensively studied, with several large-scale cancer genome analyses demonstrating a strong correlation between *TP53* mutations and aneuploidy, one of the key hallmarks of CIN [40,104,105]. These analyses, based on The Cancer Genome Atlas (TCGA) datasets, defined aneuploidy as encompassing both numerical and arm-level copy number alterations (CNAs), although some studies also included focal CNAs. Across various cancer types, the occurrence of copy number alterations was found to strongly correlate with overall mutation burden, and the mutations in *TP53* were identified as the most strongly associated with CIN [104,106]. The influence of *TP53* mutations on CIN becomes even more evident when comparing individual cancer types. Samples with mutated *TP53* consistently show a significant enrichment of copy number alterations compared to the wild-type *TP53* [40]. This highlights the pervasive impact of p53 dysfunction on genomic instability.

## 5. Early Response to Chromosome Missegregation and Whole-Genome Duplication

Chromosome missegregation occurs during mitosis when chromosomes are not properly divided between the two daughter cells. This leads to aneuploidy, a condition where cells have an abnormal, imbalanced number of chromosomes [107]. Analysis of cellular fates after chromosome missegregation or genome doubling showed that the loss of p53 function favors the survival of these cells, thus linking p53 tightly with CIN.

### 5.1. p53 in the Early Response to Chromosome Missegregation

In the context of p53 and its early response to chromosome missegregation, both the mutation type and the abundance of the mutant p53 protein in tumor cells play a critical role. Mutations in p53, particularly in variants such as R273H and G245D, have been found to predispose cells to chromosomal abnormalities. Under conditions of replication stress, such as those induced by hydroxyurea, cells with these mutant p53 forms show a higher frequency of chromosomal abnormalities than control cells [61]. This suggests that mutant p53 enables CIN and aneuploidy, and that different p53 aberrations may have different consequences. Mutant p53 proteins accumulate in cancer cells due to impaired proteasomal degradation and disrupted feedback loops (Figure 3) [108]. Overabundant mutant p53 (e.g., R175H, R248Q, and R273H) interacts widely with other proteins via its unstructured N- and C-terminal regions, thus disrupting regulatory pathways [109]. Notably, the overabundance of mutant p53 in tumors is frequently associated with additional genetic alterations, such as aneuploidy, the loss of p16INK4A and MDM2, as well as the loss of the p53 pathway [79,110].

Studies involving HCT116 cell lines found that under normal conditions, both p53-null and wild-type cells maintained a stable diploid state. However, after induced chromosome missegregation, only the p53-null cells exhibited a significant increase in aneuploidy, along with elevated EdU (5-ethynyl-2′-deoxyuridine) incorporation, which allows the labeling of newly synthesized DNA, indicating increased proliferation despite aneuploidy [111,112]. Similarly, in RPE1 cells, the loss of p53 function led to an increased occurrence of whole-chromosome gains and losses, driving aneuploidy and genomic instability [113,114]. Importantly, p53 loss permitted the survival and proliferation of aneuploid cells when missegregation occurred, rather than directly causing CIN. Those findings underscore the essential role of functional p53 in appropriate response to chromosomal errors and maintaining chromosomal stability.

Chromosome missegregation events are known to trigger a response in cells with functional p53, resulting in increased p53 levels and a subsequent delay in the cell cycle. This delay is marked by the accumulation of p21, a cyclin-dependent kinase inhibitor, which prevents the propagation of aneuploid cells. Extended mitosis can lead to the formation of 53BP1–USP28–p53 complexes, which also activate a p53 response in G1 to prevent the proliferation of progeny with mitotic errors [115]. Studies indicate that cells expressing mutant forms of p53 can proliferate even when subjected to mitotic stress or chromosomal errors, as these mutations reduce p53’s ability to induce cell cycle arrest effectively [116,117]. Further, p53 monitors chromosomal integrity through phosphorylation events during anaphase. Specifically, histone H3.3 phosphorylation at Ser31 induces lagging or misaligned chromosomes and triggers p53 activation [118]. This mechanism ensures the removal of cells with potential genomic instability from the proliferating pool.

Chromosome missegregation can lead to the activation of the DNA damage response (DDR), due to the double-strand DNA breaks (DSBs) that may arise during aberrant mitosis. These DSBs are marked by the formation of γH2AX foci even in mitotic cells, indicating a partial activation of the DDR machinery to flag DNA damage. Full DDR activation is deferred until the transition to G1, and enhances mitotic cell survival when exposed to DNA double-strand breaks [119]. The genome-protective role of the DDR depends on its ability to delay cell division until damaged DNA can be fully repaired. However, when DNA damage is induced during mitosis, the DDR unexpectedly induces chromosome errors, linking structural and numerical CIN [120]. In cells with functional p53, DNA damage leads to the phosphorylation and elevation of p53 levels, ensuring that damaged cells, including those with breaks potentially linked to errors during cytokinesis, do not proliferate [121]. Notably, DNA damage and cell cycle arrest typically occur only when a significant level of chromosome missegregation is observed, such as the missegregation of multiple chromosomes; a single or few missegregated chromosomes may not be sufficient to trigger a DDR response and arrest [113,120]. However, the exact mechanisms by which these breaks arise remain unclear. In p53-mutated cells, the DDR becomes less effective, allowing cells with significant DNA damage to evade apoptosis and continue dividing, which results in the accumulation of additional genetic abnormalities (Figure 3) [122].

The spindle assembly checkpoint (SAC) is critical for preventing chromosome segregation errors by halting the metaphase-to-anaphase transition to allow error correction. While the SAC itself does not directly detect DNA damage, p53 helps to regulate the cellular response to errors by promoting senescence or apoptosis. Vice versa, p53 is not a part of the SAC signaling pathway [113,123]. Errors resulting from abnormal mitosis can be bypassed in p53-deficient cells, allowing division to continue despite severe chromosomal defects [124,125]. During mitosis, wild-type p53 does not influence the progression from metaphase to anaphase. In contrast, mutant p53 disrupts mitotic arrest by failing to downregulate cyclin-dependent kinases regulatory subunit 1 (CKS1), which is essential in human fibroblasts. The overexpression of CKS1 in cells with mutant p53 leads to unscheduled Cyclin B degradation and polyploidy, driving CIN [126]. This disruption of mitotic control is distinct from the effects of p53 loss, which primarily leads to CIN by eliminating tumor-suppressive functions, such as DNA damage repair, cell cycle arrest, and apoptosis induction. While mutant p53 clearly contributes to CIN by allowing survival of aberrant cells, it remains unclear whether it may also promote chromosomal abnormalities.

The generation of reactive oxygen species (ROS) and subsequent oxidative stress is an early response to chromosome missegregation. For example, primary fibroblasts from aged mice exhibit increased chromosome missegregation and micronucleation correlated with mitochondrial dysfunction and elevated ROS. Antioxidant treatments have been found to reduce missegregation rates, indicating that ROS exacerbate CIN through replication stress [127]. While mild oxidative stress activates ROS-signaling pathways that protect against damage and initiate repair mechanisms, excessive stress interferes with transcription and replication processes, exacerbating genomic instability [128]. Aneuploid cells, already struggling with genomic integrity, are particularly vulnerable to elevated ROS. This early oxidative stress response highlights the importance of redox and metabolic balance in preventing tumor progression and maintaining genomic stability.

Metabolic dysfunction, including mitochondrial impairment leading to ROS generation, is also a key contributor to age-related chromosome missegregation and aneuploidy [129]. Altered metabolism, often driven by mutant p53 proteins, plays a significant role in promoting tumor development. Hot-spot mutants R175H and R273H enhance glycolysis and OxPhos in cancer cells, whereas other mutants (e.g., H179R) suppress mitochondrial metabolism. Interestingly, these metabolic effects vary depending on the cellular context, with normal cells often exhibiting contrasting metabolic responses to the same mutants (Figure 3) [130]. This suggests a heterogeneity in the metabolic GOF effects of mutant p53 proteins. Taken together, p53 and its mutant variant play a decisive role in the early response to chromosome missegregation.

### 5.2. p53 in the Early Response to Whole-Genome Doubling

According to recent estimates, nearly 40% of human tumors have undergone genome duplication during their development. The resulting tetraploidy poses a substantial oncogenic risk due to increased genomic instability, and is controlled by a p53-dependent barrier that limits the proliferation of cells with aberrant karyotypes [131,132]. A high prevalence of *TP53* mutations has been observed in cancers with WGD, particularly in early-onset colorectal cancer (EO-CRC), suggesting that the early loss of the p53 function enhances the survival of cells after WGD [133] (Figure 2B). The absence of p53 is strongly linked to increased CIN and predisposes cells to WGD, with *TP53* loss identified as an early event often preceding WGD and driving complex copy number variations (Figure 4) [134]. Additionally, recent studies have revealed that functional p53 can facilitate WGD under replication stress by activating p21 [135], while heterozygous loss of p53 alone is insufficient to enable CIN [136].

One of the earliest responses to WGD is p53-mediated G1 arrest (Figure 4) [137,138]. The arrest after WGD requires both p53 and pRB that regulate the S phase entry [137,139]. A study by Ganem et al. demonstrated that p53 depletion allowed 95.1% of examined tetraploid cells to enter the S-phase, confirming that the arrest of tetraploid cells is dependent on p53 function through p53 activation of cell cycle inhibitors such as CDKN1A/p21 [140]. Supernumerary centrosomes, often a consequence of WGD due to cytokinesis, can activate the PIDDosome complex, which in turn triggers the Caspase-2-mediated cleavage of MDM2, the stabilization of p53, and p21-dependent cell cycle arrest, serving as a first barrier to the proliferation of cells with extra centrosomes [141]. When p53 is mutated, this checkpoint fails, allowing the proliferation of tetraploid cells, which is usually associated with increased genomic instability and tumorigenic potential [139,142]. If the WGD cells manage to bypass p53-mediated G1 arrest, the Hippo pathway, in conjunction with p53, prevents tetraploid cells from bypassing the G1/S checkpoint, thereby limiting excessive cell proliferation (Figure 4) [140]. Despite strong p53-mediated arrest upon WGD, its loss alone does not lead to increased WGD or aneuploidy, indicating that CIN and tumorigenesis involve a two-step process: 1) abnormal mitosis followed by 2) a failure of G1 surveillance [139,143]. It should be noted that the precise triggers of p53 activation following WGD remain unclear, apart from the detection of supernumerary centrosomes, leaving a gap in understanding how p53 is specifically engaged in response to this karyotype aberration.

Upon WGD, cells can face a variety of outcomes, including cell cycle arrest, death, or senescence (Figure 4). Senescence is a permanent state of cell cycle arrest that serves as a protective barrier, preventing the proliferation of cells with abnormal genomic content. In the context of WGD, the tumor suppressor p53 plays a crucial role in triggering senescence [144]. However, when p53 is mutated or lost, this protective mechanism is compromised. In some cases, the Δ133p53 isoform acts as a dominant-negative regulator, inhibiting the p53-dependent expression of genes involved in the induction of senescence. As a result, this inhibition, similar to p53 deletion, allows cells to bypass senescence, increasing their potential for malignant transformation [145,146,147]. This checkpoint, however, can be bypassed by even the subtle activation of growth factor signaling or by the increased expression of cyclin D1 and D2, which can override this checkpoint and allow progression through the cell cycle [148]. Additionally, loss of the tumor suppressors such as *SPINT2*, which regulate growth factor signaling, bypass the p53 checkpoint, further promoting tumor initiation by diminishing p53-mediated control [140,149].

In conclusion, p53 plays a pivotal role in managing the consequences of WGD by inducing cell cycle arrest, senescence, and controlling metabolic stress. In its absence or when mutated, these regulatory mechanisms fail, allowing tetraploid cells to proliferate uncontrolled, significantly increasing the risk of genomic instability and cancer progression.

## 6. Late Response to Whole-Genome Duplication (WGD) and the Role of p53 Mutations

Even if cells survive the early response to WGD and continue to proliferate, their overall fitness may still be compromised due to the downstream cellular events that trigger genomic instability, tumor progression, and cancer development [150]. p53 plays a significant role in monitoring cellular integrity after WGD. WGD causes replication stress, aberrant mitoses, and disrupts chromatin organization, which leads to substantial epigenetic and transcriptional changes that fuel oncogenesis (Figure 4). These disruptions accelerate chromosomal abnormalities such as aneuploidy, which further compounds the oncogenic transformation, particularly in p53 deficient cells [151,152,153]. A notable example of this phenomenon is seen in multiple myeloma, where tumors in approximately 6% of patients exhibit tetraploid karyotypes [154]. This chromosomal abnormality is associated with a poor prognosis and is linked to *TP53* alterations [155]. Similarly, esophageal carcinoma is typically associated with whole genome doubling and *TP53* mutations [156]. These findings highlight the connection between WGD, *TP53* mutations, and aggressive malignant disease progression.

WGD in p53-mutant cells accelerates genomic instability and apoptosis evasion, driving aggressive cancer progression by allowing WGD cells to survive despite genomic instability, including high levels of aneuploidy and chromothripsis [157]. Indeed, the fraction of p53 mutated tumors nearly doubles among the WGD+ tumors compared to WGD- tumors (Figure 2B). p53 mutants interact with transcription factors such as NF-κB2, FOS, and MYC to upregulate survival pathways and inhibit apoptosis. This involves over 150 genes linked to cell survival and anti-apoptotic processes, enhancing resistance to DNA-damaging agents such as etoposide [158]. Mutant p53 also modulates miRNAs, which then leads to *STMN1* overexpression, promoting cell proliferation and apoptosis resistance [159]. This apoptosis evasion supports the selection of increasingly aggressive, tumor-promoting phenotypes, as unstable yet viable WGD cells become more adaptable to oncogenic conditions. Together, these mechanisms foster a cycle of instability, chemoresistance, and proliferation, advancing cancer evolution.

WGD-induced stresses typically activate metabolic pathways regulated by p53, promoting metabolic adaptation to maintain cellular homeostasis and preventing tumorigenesis (Figure 4). Post-WGD, altered metabolic states arise in cancer cells carrying p53 mutation due to p53’s multifaceted role in energy regulation. Normally, p53 balances glycolysis and oxidative phosphorylation, thus suppressing tumorigenesis and the Warburg effect by modulating genes such as *SCO2*, *GLS2*, *TIGAR*, and *GLUT* transporters [19,160,161]. This regulation promotes oxidative phosphorylation while selectively limiting glycolysis, reducing reactive oxygen species, and enhancing antioxidant defense. In the context of WGD, these metabolic adaptations may help tetraploid cells to cope with the increased metabolic demands and oxidative stress associated with their doubled genome. The control of these pathways by mutated p53 is compromised, often resulting in metabolic shifts that enhance glycolytic activity, support uncontrolled proliferation, and alter cellular responses to oxidative stress, further promoting the survival and expansion of WGD cells.

Mutant p53 serves as a significant carcinogenic factor in post-WGD cancer cells by driving chemoresistance through multiple mechanisms (Figure 4). First, mutp53 promotes drug efflux by activating the expression of the MDR1 (multiple drug resistance) and enhancing the expression of ATP-binding cassette (ABC) transporters that expel chemotherapeutic agents, leading to multidrug resistance [162]. It also increases the metabolism of drugs by upregulating cytochrome P450 enzymes, such as CYP3A4, resulting in reduced efficacy of several chemotherapeutics [163]. Moreover, mutated p53 (R273H and R273G) enhances DNA repair pathways by upregulating key genes such as *BRCA1*, which increases resistance to DNA-damaging agents such as cisplatin and etoposide [164]. Autophagy inhibition further contributes to chemoresistance by preventing cell death, as cytoplasmic aggregation of mutated p53 inhibits autophagic processes [165]. In addition, mutant p53 influences the tumor microenvironment by enhancing integrin and EGFR signaling through the Rab-coupling protein. This sustained activation of *EGFR* and integrin pathways may support the survival and proliferation of WGD cells, enabling them to thrive despite the challenges of genomic instability [166].

## 7. Conclusions

The interplay between chromosome missegregation, WGD, and p53 function is central to understanding genomic instability and its role in cancer progression. Chromosome missegregation and WGD are key events that disrupt genomic stability, leading to aneuploidy and polyploidy—hallmarks of many cancers. While functional p53 safeguards cellular integrity through robust surveillance and repair mechanisms, its loss or mutation creates a permissive environment for genomic instability and cancer progression. This role of p53 as both a protector against and enabler of chromosomal abnormalities highlights its importance in cancer biology.

Functional p53 mitigates CIN by activating pathways that respond to the chromosomal errors and initiate corrective actions or eliminate defective cells. In contrast, mutations in p53 profoundly alter this protective response. Mutant p53 variants, such as R273H and R175H, not only fail to arrest the cell cycle in response to chromosomal errors but also actively promote the proliferation of cells with chromosomal abnormalities. These GOF mutants promote oncogenic behaviors by enabling cells to evade negative consequences of CIN. However, it remains an unresolved question whether mutant GOF p53 only enables survival of cells with chromosomal abnormalities or whether they could also contribute to their generation. Additionally, it remains unclear whether there are significant differences between the effects of GOF mutations and complete p53 loss. These gaps in understanding highlight critical areas for further investigation.

WGD represents a distinct route to genomic instability and typically leads to aneuploidy as cells undergo subsequent divisions, resulting in cells with amplified oncogenic potential. The loss or mutation of p53 allows tetraploid cells to evade arrest and progress toward malignancy. Additionally, p53 mutations correlate with early WGD events, enabling these cells to survive despite increased CIN. Furthermore, the interplay between p53 and other regulatory pathways, such as the Hippo pathway, highlights the multifaceted mechanisms through which p53 controls cell fate after WGD. When these pathways are disrupted, either through p53 loss or other oncogenic alterations, cells gain a proliferative advantage despite genomic instability, accelerating cancer progression. Mutant p53 further exacerbates genomic instability by modulating key metabolic pathways, favoring glycolysis over oxidative phosphorylation, and enabling adaptation to oxidative stress. This metabolic shift supports the energy demands of rapidly proliferating, genetically unstable cancer cells and provides potential targets for therapeutic intervention.

Despite substantial progress, critical questions remain regarding the role of p53 in the context of chromosome missegregation and WGD. Most experimental studies have primarily focused on p53 deletions rather than mutations, leaving the specific roles of these mutations underexplored. This will require further investigation into the distinct mechanisms by which p53 deletions and mutations influence chromosomal stability and tumor evolution. Future studies should also explore therapeutic strategies that exploit the vulnerabilities of p53-mutant cells, such as targeting metabolic reprogramming or enhancing alternative DDR pathways. This could provide new approaches for combating the large fraction of cancers characterized by high genomic instability and p53 mutations.

## Figures and Tables

**Figure 1 biomolecules-15-00244-f001:**
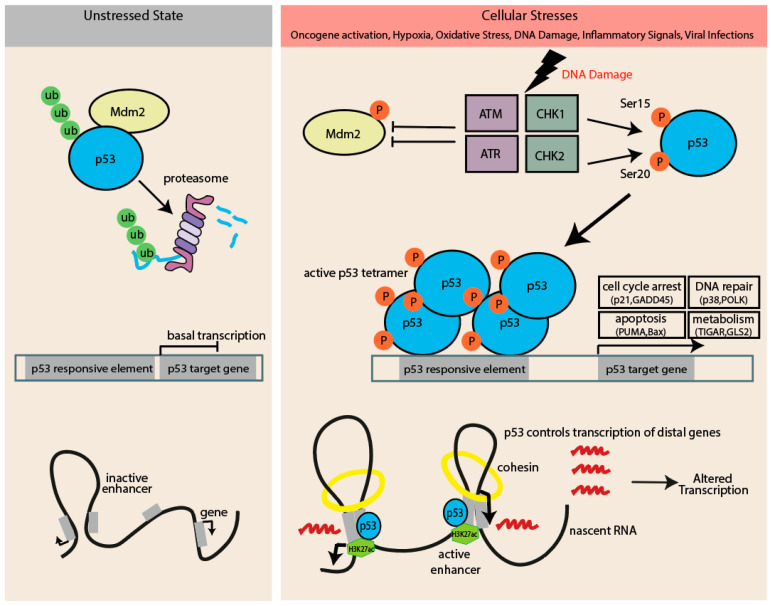
Regulation of p53 protein levels and activity in unstressed and stressed cellular states. Models depict the mechanisms that regulate p53 protein levels and activity in unstressed cells and in cells experiencing stress upon DNA damage. p53 is activated by a range of cellular stress signals, including activation of oncogenes, hypoxic conditions, DNA damage, and oxidative stress from reactive oxygen species (ROS). p53 also controls transcription of distal genes by promoting enhancer activation, marked by increased H3K27ac and nascent RNA, and facilitating cohesin-mediated DNA. (Ub, ubiquitin; P, phosphorylation).

**Figure 4 biomolecules-15-00244-f004:**
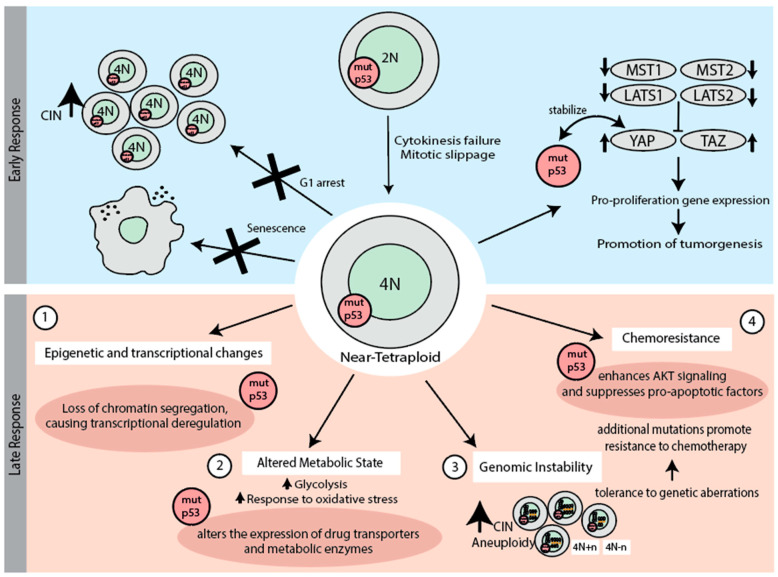
The role of mutant p53 (mutp53) on cellular responses to whole-genome doubling (WGD), highlighting early and late responses. In the early response, mutant p53 disrupts G1 arrest, enables evasion of senescence, and inhibits the Hippo pathway, promoting pro-proliferative gene expression. In the late response, (1) mutp53 drives epigenetic and transcriptional changes through loss of chromatin segregation (LCS) and genomic destabilization, resulting in transcriptional deregulation and epigenetic vulnerabilities that favor tumor development. (2) Mutp53 also alters metabolic states by modifying the expression of drug transporters and metabolic enzymes, such as MDR1, affecting chemotherapeutic agent metabolism. (3) Furthermore, mutp53 enhances chemoresistance by activating AKT signaling and suppressing pro-apoptotic factors like BMF, contributing to resistance and cancer cell survival. (4) Additionally, mutp53 promotes genomic instability by impairing DNA repair, leading to the accumulation of deleterious structural variants and enabling the proliferation of WGD cells despite extensive genomic alterations.

**Table 1 biomolecules-15-00244-t001:** Common *TP53* missense mutations associated with chromosomal instability.

Mutation Locus	Activity Status	Activity Change	Phenotype	Associated Cancer Type	References
G245S	LOF	Destabilizes p53 protein and impairs DNA-binding ability	Exhibits altered conformational dynamics, reduced flexibility in loop L3, increased genomic instability, and resistance to chemotherapeutic agents.	Not specified	[47]
R175H	GOF	Modulates gene expression by binding to promoters, either independently or with transcription factor assistance.	Increases tumorigenicity by driving oncogenic pathways, proliferation, drug resistance, inflammation, angiogenesis, and metabolic reprogramming.	Key cancer types: Colorectal, Breast, Lung, Gastric, Endometrial, Pancreatic	[48,49]
R273H	GOF	Loss of tumor-suppressive functions while acquiring new oncogenic properties.	Enhances survival, migration, invasion, and chemoresistance by suppressing miR-27a, increasing EGFR, and activating ERK1/2 to drive proliferation and tumor growth.	Key cancer types: Colorectal, Breast, Lung, Head and Neck Squamous Cell Carcinoma	[50,51,52]
R248Q	DN	Interferes with wild-type p53 functions through dominant-negative effects.	Increased motility and invasiveness, coupled with inhibition of macro autophagy.	Key cancer types: High-Grade Serous Ovarian Carcinoma, Colorectal, Breast, Lung, Pancreatic	[53,54,55,56]
R249S	GOF LOF	Promotes proliferation via c-MYC-dependent ribosomal biogenesis but fails to bind p53 response elements.	Increased genomic instability, including interchromosomal translocations and aneuploidy.	Key cancer types: Hepatocellular carcinoma (HCC) (30% of all TP53 mutations in HCC), Lung, Head and Neck Squamous Cell Carcinoma, Colorectal	[57,58]
D281G	GOF	Activates cell cycle and survival genes while inducing p53 instability and aggregation.	Enhanced growth, survival, and angiogenesis, with increased metastasis, therapy resistance, and potential structural instability.	Key cancer types: Lung, Breast, Colorectal	[54,59,60]

GOF: Gain of Function, LOF: Loss of Function, DN: Dominant Negative.

## Data Availability

Not applicable.

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
