# Peer review of "The Role of p53 Mutations in Early and Late Response to Mitotic Aberrations"

_biomolecules, 2025, doi:10.3390/biom15020244_

Round 1
Reviewer 1 Report
Comments and Suggestions for Authors
The role of p53 mutations in early and late response to mitotic aberrations
By Anna-Luisa Hertel and Zuzana Storchova
Major points:
It is very good how the authors systematically described the many mutants and corresponding phenotypes of p53; concise and informative. Overall, the paper was well written.
I’m not entirely convinced from this paper that p53 mutations cause WGD, allow WGD or are associate with WGD. p53 is responding to mitotic events, not causing them.
The definitions of CIN and aneuploidy seem problematic.
CIN is chromosome instability. Aneuploidy is an abnormal number of chromosomes, either missing or extra. Aneuploidy is a consequence of CIN. Chromosome missegregation causes CIN, and CIN results in aneuploidy.
Dr. Bert Vogelstein’s and Dr. Christoph Lengauer’s groups showed that cells completely lacking p53 did not exhibit CIN, thus not aneuploidy, although a slight tendency toward tetraploidization was observed(1).
1. F. Bunz et al., Targeted inactivation of p53 in human cells does not result in aneuploidy. Cancer Res 62, 1129-1133 (2002).
The authors conclude that loss of p53 leads to WGD, and WGD often leads to CIN, which is fine. However, to avoid misleading the reader, it is important to emphasize that loss of p53 does not directly cause chromosome missegregation or CIN.
Minor points:
Line 139 Typo: not “base pairs”, should read “amino acids”
Line 149 says that GOF mutant p53 does not bind DNA directly but Line 156 says that it does.
Line 234 Figure 2 seems to indicate that WT p53 cells have ~5-8% WGD, implying that WGD occurs before p53 mutations. Why doesn’t the total of WT WGD and non-WGD = 100%?
Line 156 Typo: “disrupts” should read “disrupt”
Line 279 the word “indicates” is too conclusive. “significantly associated” is not causation.
Line 305-308 p53 null cells did not induce CIN without promotion of missegregation. This statement contradicts the claim that p53 is driving CIN.
Line 322 typo: missing verb? “Induces”?
Line 422 typo: apart “from” the
Line 433 typo: however, "can" be
Line 447 typo? “Instability” instead of stability? Or Freudian slip?
Line 448 Which mutants produced WGD?
Line 450 typo: The use of “this” usually refers to the previous noun, but there are several to choose from in the previous sentence.
Author Response
Reviewer #1
Comments and Suggestions for Authors
The role of p53 mutations in early and late response to mitotic aberrations
By Anna-Luisa Hertel and Zuzana Storchova
Major points:
It is very good how the authors systematically described the many mutants and corresponding phenotypes of p53; concise and informative. Overall, the paper was well written.
>> We thank the reviewer for the positive evaluation and for the constructive remarks, which we addressed in the revised version.
I’m not entirely convinced from this paper that p53 mutations cause WGD, allow WGD or are associate with WGD. p53 is responding to mitotic events, not causing them.
>> We agree with this interpretation and in fact, we never claimed that p53 mutations cause WGD, only that they are more frequent in cells that underwent WGD. We have now adjusted the sentence where both interpretations were possible (see below).
The definitions of CIN and aneuploidy seem problematic.
CIN is chromosome instability. Aneuploidy is an abnormal number of chromosomes, either missing or extra. Aneuploidy is a consequence of CIN. Chromosome missegregation causes CIN, and CIN results in aneuploidy.
>> We agree with this interpretation, which we clearly state in the manuscript. We have now reinforced this statement to avoid possible misinterpretation.
Dr. Bert Vogelstein’s and Dr. Christoph Lengauer’s groups showed that cells completely lacking p53 did not exhibit CIN, thus not aneuploidy, although a slight tendency toward tetraploidization was observed(1).
- Bunzet al., Targeted inactivation of p53 in human cells does not result in aneuploidy. Cancer Res 62, 1129-1133 (2002).
>> We are well aware of this paper, which we cited in our manuscript. We now reinforced the statements to avoid possible misinterpretations.
The authors conclude that loss of p53 leads to WGD, and WGD often leads to CIN, which is fine. However, to avoid misleading the reader, it is important to emphasize that loss of p53 does not directly cause chromosome missegregation or CIN.
>> We agree with interpretation, we have refenced the Vogelstein paper already in the first version. To avoid any confusion, we carefully checked all statements in the review concerning p53 and CIN relationship. We made sure that it is clear that a mutation in p53 alone does not lead to CIN.
Minor points:
Line 139 Typo: not “base pairs”, should read “amino acids”
>> We corrected this typo.
Line 149 says that GOF mutant p53 does not bind DNA directly but Line 156 says that it does.
>> We now corrected the sentences so that it is clear that the association of the GOF mutant p53 binds the promotors indirectly, through association with other proteins.
Line 234 Figure 2 seems to indicate that WT p53 cells have ~5-8% WGD, implying that WGD occurs before p53 mutations. Why doesn’t the total of WT WGD and non-WGD = 100%?
>> We thank the reviewer for noticing this discrepancy. Yes, indeed, in 5 – 8% of cancer the whole genome doubling occurs independently of p53. It is possible that the functionality of the p53 pathways is disrupted in these cells through other mutations, but the p53 is not mutated. The total of WGD+ and WGD- does not reach 100 %, because in some cases the data do not allow to clearly define the WGD status. We now clarified this aspect in the figure legend.
Line 156 Typo: “disrupts” should read “disrupt”
>> We corrected this typo.
Line 279 the word “indicates” is too conclusive. “significantly associated” is not causation.
>> We replaced the word “indicates” with the word “suggest”.
Line 305-308 p53 null cells did not induce CIN without promotion of missegregation. This statement contradicts the claim that p53 is driving CIN.
>> When stating that “p53 mutation is driving CIN”, we do not mean that it triggers missegregation, but that it allows survival of cells with aberrant chromosomal content. In that sense, p53 mutations indeed are driving CIN, because without p53 mutations the CIN cells would not survive. However, we understand reviewer’s point and we now carefully checked the manuscript and corrected the sentences which would possibly allow the interpretation that p53 mutation is a cause of chromosome missegregation.
Line 322 typo: missing verb? “Induces”?
>> We corrected this typo.
Line 422 typo: apart “from” the
>> We corrected this typo.
Line 433 typo: however, "can" be
>> We corrected this typo.
Line 447 typo? “Instability” instead of stability? Or Freudian slip?
>> We corrected this typo.
Line 448 Which mutants produced WGD?
>> We are not sure to which point this comment refers to.
Line 450 typo: The use of “this” usually refers to the previous noun, but there are several to choose from in the previous sentence.
>> We specified which noun is relevant in this context.
Reviewer 2 Report
Comments and Suggestions for Authors
The present manuscript is a review related to the role of TP53 mutations in early and late response to mitotic aberrations. It summarizes the current insights of this gene in protecting cells from chromosome copy number alterations and discusses the implications of TP53 dysfunction for the adaption and propagation of cancer cells. The relationship between TP53 mutations and chromosome instability (CIN), marked by changes in chromosome number and structure, as well as whole-genome doubling, a process that facilitates CIN onset and tolerance, and their association with poor patient prognosis in multiple cancer types is analyzed. The manuscript was competently performed, the subject was consistently developed but some aspects must be considered.
Major comments
- Page 4, Lines 130-132. The authors refer biallelic TP53 mutations as common in aggressive multiple myeloma and mention Ref. 38 (Agarwal, H., Cell Death Differ, 2024. 31(7): 855-867), that is not related to the subject.
Biallelic TP53 mutations correspond to double hit multiple myelomas, a subgroup that present complete TP53 inactivation due to mutation in both alleles or deletion in one and mutation in the other (Walker et al, Leukemia 2019; 33: 159-170; Martello et al, Blood Cancer J 2022; 12: 15). In addition, the presence of TP53 deletion or mutation also confer poor prognosis to multiple myeloma patients (Corre et al. Blood. 2021;137: 1192-5). This point must be corrected in the text.
- Page 6, Lines 256-257 and Page 7, Lines 258-260. The authors refer the prognostic significance of deletion 17p in acute myeloid leukemia and non-small cell lung cancer. It is also important to refer the prognostic and predictive value of TP53 del/mut. in chronic lymphocytic leukemia, which study is mandatory before starting treatment (Campo et al, Haematologica 2018; 103: 1956-68; Bomben et al. Clin Cancer Res 2021; 27: 5566-75; Malcikova et al. Leukemia 2024; 38: 1455-68). A commentary about it must be introduced in the manuscript.
- Page 11, Late Response to Whole-Genome Duplication (WGD) and the Role of p53 Mutations. In this point, it would be interesting that the authors introduce in the manuscript the example of multiple myeloma patients with tetraploidy karyotypes and their relation to poor outcome. This alteration was observed in about 6% of newly diagnosed multiple myeloma cases (Sidana et al. Am J Hematol 2019; 94: E117-E120; doi: 10.1002/ajh.25420) as well as in 10% of relapsed patients (Locher M, et al. Eur J Haematol. 2023;110:296-304) in both situations associated to TP53 alterations.
Table 1.
- The second column “Type” must be deleted and the condition “Missense in DNA binding domain” introduced in the title. It is not necessary a column that repeat the same in all lines.
- Please, clarify the abbreviations of the column “Activity status” at the bottom of the table.
- The references present in the table are not detailed. Please, detail them below the table in a Reference section.
Minor comments
- The nomenclature of genes and proteins must be revised along the manuscript. According to the established, human gene symbols are italicized, with all letters in uppercase [e.g., BCL2 (BCL2 Apoptosis Regulator)] and protein designations are the same as the gene symbol except that they are not italicised (e.g. BCL2) https://www.genecards.org/cgi-bin/carddisp.pl?gene=BCL2 https://www.pnas.org/doi/10.1073/pnas.2025207118
Genes are correctly written along the text but not proteins (e.g., Mdm2 must be written MDM2). Please correct along the text and figures.
- Page 8, Line 308: Please clarify the abbreviation EdU
Author Response
Reviewer # 2
Submission Date
20 December 2024
Date of this review
05 Jan 2025 12:40:39
The present manuscript is a review related to the role of TP53 mutations in early and late response to mitotic aberrations. It summarizes the current insights of this gene in protecting cells from chromosome copy number alterations and discusses the implications of TP53 dysfunction for the adaption and propagation of cancer cells. The relationship between TP53 mutations and chromosome instability (CIN), marked by changes in chromosome number and structure, as well as whole-genome doubling, a process that facilitates CIN onset and tolerance, and their association with poor patient prognosis in multiple cancer types is analyzed. The manuscript was competently performed, the subject was consistently developed but some aspects must be considered.
>> We thank the reviewer for the positive evaluation and for the constructive remarks, which we addressed in the revised version.
Major comments
- Page 4, Lines 130-132. The authors refer biallelic TP53 mutations as common in aggressive multiple myeloma and mention Ref. 38 (Agarwal, H., Cell Death Differ, 2024. 31(7): 855-867), that is not related to the subject.
>> We thank the reviewer for noticing the disparity. We have now included the correct reference (Shahzad et al, 2024).
Biallelic TP53 mutations correspond to double hit multiple myelomas, a subgroup that present complete TP53 inactivation due to mutation in both alleles or deletion in one and mutation in the other (Walker et al, Leukemia 2019; 33: 159-170; Martello et al, Blood Cancer J 2022; 12: 15). In addition, the presence of TP53 deletion or mutation also confer poor prognosis to multiple myeloma patients (Corre et al. Blood. 2021;137: 1192-5). This point must be corrected in the text.
>> We thank the reviewer to make us aware of this point. We have added this to the manuscript. At the same time, we did not want to focus on one tumor type only.
- Page 6, Lines 256-257 and Page 7, Lines 258-260. The authors refer the prognostic significance of deletion 17p in acute myeloid leukemia and non-small cell lung cancer. It is also important to refer the prognostic and predictive value of TP53 del/mut. in chronic lymphocytic leukemia, which study is mandatory before starting treatment (Campo et al, Haematologica 2018; 103: 1956-68; Bomben et al. Clin Cancer Res 2021; 27: 5566-75; Malcikova et al. Leukemia 2024; 38: 1455-68). A commentary about it must be introduced in the manuscript.
>> We thank the reviewer to make us aware of these additional studies. The review focuses broadly on the role of p53 in cellular response to WGD and aneuploidy in the context of cancer, and therefore we did not wish to discuss in greater detail one particular cancer type. We have now added a brief text clarifying these aspects.
- Page 11, Late Response to Whole-Genome Duplication (WGD) and the Role of p53 Mutations. In this point, it would be interesting that the authors introduce in the manuscript the example of multiple myeloma patients with tetraploidy karyotypes and their relation to poor outcome. This alteration was observed in about 6% of newly diagnosed multiple myeloma cases (Sidana et al. Am J Hematol 2019; 94: E117-E120; doi: 10.1002/ajh.25420) as well as in 10% of relapsed patients (Locher M, et al. Eur J Haematol. 2023;110:296-304) in both situations associated to TP53 alterations.
>> We thank the reviewer to make us aware of these additional studies. The review focuses broadly on the role of p53 in cellular response to WGD and aneuploidy in the context of cancer, and therefore we did not wish to discuss in greater detail one particular cancer type. We also wish to keep the focu sbalanced and show examples from different cancer types. We have now added a brief text clarifying these aspects.
Table 1.
- The second column “Type” must be deleted and the condition “Missense in DNA binding domain” introduced in the title. It is not necessary a column that repeat the same in all lines.
>> We thank the reviewer for the suggestion how to simplify the information-rich table.
- Please, clarify the abbreviations of the column “Activity status” at the bottom of the table.
>> We thank the reviewer for the suggestion how to clarify the table.
- The references present in the table are not detailed. Please, detail them below the table in a Reference section.
>> We now combined the table with the main text and the references are included in the main referencing.
Minor comments
- The nomenclature of genes and proteins must be revised along the manuscript. According to the established, human gene symbols are italicized, with all letters in uppercase [e.g., BCL2 (BCL2 Apoptosis Regulator)] and protein designations are the same as the gene symbol except that they are not italicised (e.g. BCL2) https://www.genecards.org/cgi-bin/carddisp.pl?gene=BCL2 https://www.pnas.org/doi/10.1073/pnas.2025207118
Genes are correctly written along the text but not proteins (e.g., Mdm2 must be written MDM2). Please correct along the text and figures.
>> We corrected the nomenclature es requested.
- Page 8, Line 308: Please clarify the abbreviation EdU
>> We explained the abbreviation.
Round 2
Reviewer 2 Report
Comments and Suggestions for Authors
Most of the comments have been satisfactorily answered, except for the next paragraph.
Page 4, second paragraph, lines 132-135. Biallelic TP53 mutations, including mutations either in both alelles, or a deletion in one and mutation in the other, are common in aggressive cancers such as multiple myeloma. This mutations lead to complete TP53 inactivation, and correlate with poor prognosis [40-44].
Biallelic TP53 inactivation is not common in multiple myeloma. It corresponds to 6% of patients. Multiple myeloma is not the best example for this affirmation. Other types of cancers must be referred. Please modified the text.
Author Response
Reviewer #2, second round of reviews
Page 4, second paragraph, lines 132-135. Biallelic TP53 mutations, including mutations either in both alelles, or a deletion in one and mutation in the other, are common in aggressive cancers such as multiple myeloma. This mutations lead to complete TP53 inactivation, and correlate with poor prognosis [40-44].
Biallelic TP53 inactivation is not common in multiple myeloma. It corresponds to 6% of patients. Multiple myeloma is not the best example for this affirmation. Other types of cancers must be referred. Please modified the text.
>>We agree that multiple myeloma is not the most prominent example of bi-allelic mutations in cancer, but it was our understanding that the reviewer requested in the previous round of revisions to add this information. We have now amended the text to more general statement that biallelic p53 mutations are frequent in aggressive tumors, as it was stated in the first version. We hope that this addresses reviewer’s concern.